# Cache me if you Can: an Online Cost-aware Teacher-Student Framework to Reduce the Calls to Large Language Models

**Ilias Stogiannidis**[1,2] **Stavros Vassos**[2] **Prodromos Malakasiotis**[1,4] **Ion Androutsopoulos**[1,3]

[1]Department of Informatics, Athens University of Economics and Business, Greece

[2]Helvia.ai

[3]Archimedes Unit, Athena Research Center, Greece

[4] Workable

{stoyianel, rulller, ion}@aueb.gr, stavros@helvia.ai

## Abstract

Prompting Large Language Models (LLMs) performs impressively in zero- and few-shot settings. Hence, small and medium-sized enterprises (SMEs) that cannot afford the cost of creating large task-specific training datasets, but also the cost of pretraining their own LLMs, are increasingly turning to third-party services that allow them to prompt LLMs. However, such services currently require a payment per call, which becomes a significant operating expense (OpEx). Furthermore, customer inputs are often very similar over time, hence SMEs end-up prompting LLMs with very similar instances. We propose a framework that allows reducing the calls to LLMs by caching previous LLM responses and using them to train a local inexpensive model on the SME side. The framework includes criteria for deciding when to trust the local model or call the LLM, and a methodology to tune the criteria and measure the tradeoff between performance and cost. For experimental purposes, we instantiate our framework with two LLMs, GPT-3.5 or GPT-4, and two inexpensive students, a $k$-NN classifier or a Multi-Layer Perceptron, using two common business tasks, intent recognition and sentiment analysis. Experimental results indicate that significant OpEx savings can be obtained with only slightly lower performance.

## 1 Introduction

Prompting pre-trained Large Language Models (LLMs) aligned to follow instructions (Ouyang et al., 2022; Köpf et al., 2023) performs impressively well in zero- and few-shot settings. Hence, small and medium-sized enterprises (SMEs) that cannot afford the cost of creating large task-specific training datasets for model fine-tuning, but also the cost of pretraining their own LLMs, are increasingly turning to third-party services that allow them to prompt LLMs. For example, SMEs that provide customer support chatbots prompt LLMs like GPT-4 (OpenAI, 2023) to detect user intents

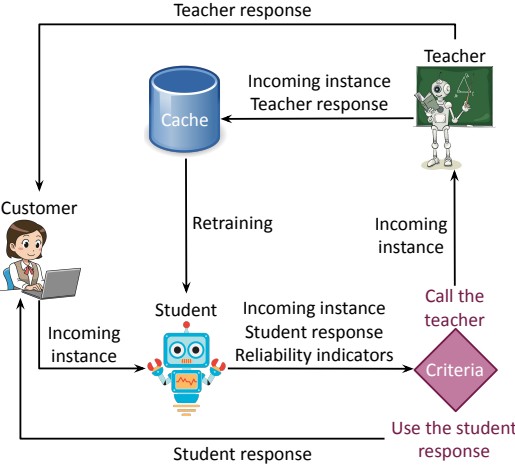

Figure 1: OCaTS architecture.

and drive the chatbot-customer interaction (Ham et al., 2020). The best LLMs, however, currently require a payment per prompting call, and these payments become a significant operating expense (OpEx) for SMEs. Furthermore, customer inputs (e.g., dialog turns) are often very similar over time, hence SMEs end up calling LLMs to handle inputs that may be very similar to inputs already handled by the LLMs in previous (already paid) calls.

We introduce the *Online Cost-aware Teacher-Student* (OCaTS) framework that allows reducing the calls to a commercial LLM, treated as a teacher model, by caching its previous responses and using them to train a local inexpensive student model. OCaTS includes criteria for deciding when to trust the student or call the teacher, and a methodology to tune the criteria and measure the tradeoff between performance and cost. Unlike common teacher-student training for knowledge distillation (Hinton et al., 2015; Gou et al., 2021), here the teacher does not train the student on all the available instances (in our case, all the incoming customer inputs). Also, unlike teacher-student approaches to self-training (Mi et al., 2021; Li et al., 2021), the teacher is already reasonably effective (but expensive). In that sense, our work is closer to ac-

tive learning (Settles, 2012; Monarch, 2021), but OCaTS trains the student on labels provided by a teacher LLM, not humans, and there is initially no large pool of unlabeled instances (customer inputs) to select from, as instances arrive online.

OCaTS can be used with any service that allows prompting LLMs, and any kind of local student model. For experimental purposes, we instantiate OCaTS with GPT-3.5 or GPT-4 as the teacher, and a $k$-NN or Multi-Layer Perceptron (MLP) classifier as the student, using an intent recognition dataset from the banking domain or a sentiment analysis dataset. Experimental results indicate that significant OpEx savings can be obtained with only slightly lower performance. For example, the $k$-NN student can handle approximately two-thirds of the incoming instances (customer inputs) of the intent recognition task without calling the GPT-4 teacher (Fig. 2, left, red line) for a decrease of less than 0.5 percentage points in accuracy (Fig. 2, middle, red and black lines). OCaTS introduces discounted versions of common evaluation measures (e.g., accuracy) that allow an SME to quantify how much it prefers to lean towards fewer calls or less user frustration (different $\lambda$ values in Fig. 2).

Our main contributions are: (i) We introduce a general teacher-student framework that helps SMEs reduce the prompting calls to commercial LLMs and the corresponding OpEx costs by caching the responses of the LLMs and training inexpensive local student models. (ii) We introduce discounted versions of common evaluation measures that allow the SMEs to quantify how much they prefer fewer LLM calls vs. increased user frustration (e.g., caused by lower accuracy) and tune the framework's criteria that decide when to trust the local student model or call the LLM teacher accordingly. (iii) We instantiate the framework with GPT-3.5 or GPT-4 as teachers, and a $k$-NN or MLP classifier as students. (iv) We perform experiments on two well-known tasks for SMEs, intent recognition and sentiment analysis, and show that significant cost savings can be obtained with only slightly lower performance. This is a first step towards exploring the benefits of the proposed framework with more datasets, models, and business scenarios.

## 2 Framework

**Architecture:** The proposed framework (OCaTS) consists of three main components (Fig. 1): a *teacher*, typically a resource-intensive model of-

fering premium results; a *student*, a cost-effective model that is typically much smaller and simpler than the teacher; a *cache*, a repository of incoming instances (e.g., customer requests) that have already been processed by the teacher. We assume that the framework is employed to handle a task for which there is no available large dataset for supervised training, apart from a few incoming instances (possibly a handful per class) annotated with the ground truth (e.g., correct labels). This is a very common case for SMEs that cannot afford the cost of creating large task-specific training datasets, but can easily construct small numbers of demonstration instances. The teacher-student setting is *online*, as every incoming instance is handled at inference time as follows. First, the student is called to handle the instance. Then some student- and task-specific *criteria*, which assess the reliability of the student's output, indicate if the student's output (e.g., label) should be used or if the teacher should be consulted. If the student's output is selected, it is returned as the response to the incoming instance. Otherwise, the teacher is called to handle the instance. In the latter case, the instance along with the teacher's result are stored in the cache. Depending on the type of student, periodic re-training takes place, to update the student with the cached instances.

**Instantiations:** In the experiments of this paper, we instantiate OCaTS with a GPT-3.5 or GPT-4 teacher, a distance-weighted $k$-NN or MLP classifier as the student, for a single-label classification task (intent recognition or sentiment analysis). In all cases, we represent each incoming instance (customer request) by its MPNet-based (Song et al., 2020) vector representation (text embedding) and we use two criteria (Fig. 1) to decide when to use the student's response or invoke the teacher: (i) the entropy of the probability distribution (over the label set) produced by the student ($k$-NN or MLP) for the incoming instance, and (ii) the distance of the vector representation of the incoming instance from the centroid of the vector representations of the $k$ most similar cached instances. Consult Nguyen et al. (2022) for other possible criteria. We leave other instantiations of OCaTS (other teachers, students, tasks, representations) for future work.

**Discounted evaluation measures:** The main goal of the proposed architecture is to reduce the number of calls to the expensive teacher model by caching previous teacher responses and using them to train a local inexpensive student model on the SME side.

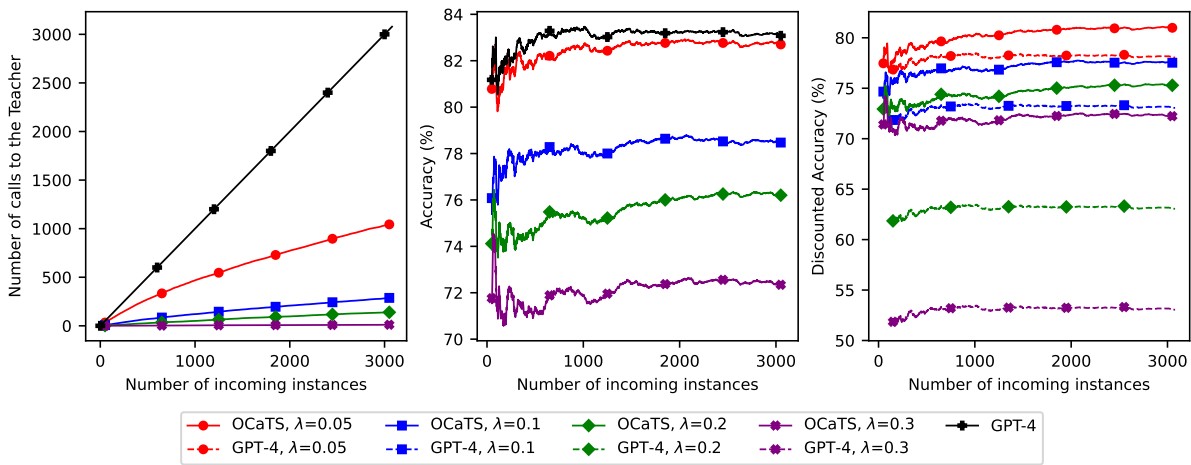

Figure 2: Number of calls to the teacher (left), accuracy (middle), discounted accuracy (right), using a GPT-4 teacher and a $k$-NN student, for various $\lambda$ values, on Banking77 data. The larger the $\lambda$ the more the SME prefers fewer calls at the expense of increased user frustration. Dashed lines show the discounted accuracy when calling GPT-4 for all incoming instances. OCaTS has a better discounted accuracy than always calling the GPT-4 teacher.

This introduces a tradeoff between the OpEx cost of calling the teacher and the frustration of the end-users when the less accurate student model is used instead. To quantify this tradeoff, we introduce a *discounted* variant $\hat{\phi}$ of any common evaluation measure $\phi$ (e.g., accuracy, F1), as follows:

$$\hat{\phi} = \phi - \lambda \cdot \frac{M}{N} = \phi - \lambda \cdot \rho, \qquad (1)$$

where $N$ is the number of incoming instances that have been processed (on which $\phi$ is measured), $M$ is the number of calls made to the teacher while processing the $N$ instances, $\rho = \frac{M}{N}$ shows for what percentage of the incoming instances we call the teacher, and $\lambda$ is a scalar specifying how intensively the measure should be discounted. Assume, for example, that the accuracy of the teacher-student combination is $\phi = 0.8$, but that this accuracy is achieved with $\rho = \frac{1}{3}$. If the SME considers this $\rho$ value (which would translate, e.g., to a monthly cost) as costly as a loss of five percentage points of accuracy, then $\hat{\phi} = 0.75$, and Eq. 1 becomes $0.75 = 0.8 - \lambda \cdot \frac{1}{3}$, from which we obtain $\lambda = 0.15$. Larger (or smaller) $\lambda$ values correspond to cases where the SME considers the same $\rho$ value more (or less) costly in terms of loss of accuracy points. We can also reformulate Eq. 1 as $\delta = \lambda \cdot \rho$, where $\delta = \phi - \hat{\phi}$ shows how much $\phi$ gets discounted to account for the cost of $\rho$. Then $\lambda$ can intuitively be thought of as a currency exchange rate, showing how expensive $\rho$ is in terms of $\delta$ (e.g., loss of accuracy in percentage points).[1]

---

[1]We implicitly assume that the exchange rate $\lambda$ is constant for all the values of $\delta$ and $\rho$. In practice, it may be different for different ranges of $\delta$ and $\rho$, but we leave this for future work.

## 3 Main experiments

Here we discuss the experiments we conducted with the GPT-4 teacher, the $k$-NN student, and the banking intent recognition dataset. In the Appendix, we report two additional sets of experiments, one where we replaced the $k$-NN student by an MLP (Appendix E) keeping the rest of the setup unchanged, and one where we replaced the task/dataset (by sentiment analysis) and the teacher (by the cheaper GPT-3.5) otherwise keeping the setup of the initial experiments (Appendix F). The additional experiments verify the conclusions of the experiments of this section.

**Intent recognition dataset:** In this section, we use Banking77 (Casanueva et al., 2020), an intent recognition dataset from the banking customer service domain. It includes 13,083 customer messages. The ground truth assigns to each message a single label (intent) from the 77 available. The dataset is divided into training (10,003 instances) and test (3,080) subsets. Appendix A shows more statistics.

**Few-shot training and development sets:** Assuming that an SME can only afford to construct a small number of training instances per class, we use only $3 \times 77 = 231$ instances from the original training set of Banking77, three per class, as a few-shot version of the training set. The 231 instances were manually selected to avoid unclear cases, e.g., similar instances with different ground truth labels. Similarly, we created a few-shot development set of $13 \times 77 = 1,001$ instances from the original training set, for hyperparameter tuning.

**Incoming instances and evaluation measure:** We use the original test set of Banking77 as the incoming instances. We repeat each experiment with five random shufflings of the test set (to obtain five different streams of input instances) and report average scores over the shufflings. We set $\phi$ to accuracy, since the test set is balanced (Appendix A).

**Teacher:** In this section, we use GPT-4 (OpenAI, 2023) as the teacher, the most capable LLM for few-shot in-context learning tasks at the time. Each prompt includes instructions, demonstrators (in-context few-shot examples), and the incoming instance to be classified; see Appendix B for details.

**Student:** In this section, a distance-weighted $k$-NN classifier is used as the student. Vector representations of the incoming instances are generated with a Sentence-Transformer (Reimers and Gurevych, 2019) variation of MPNet (Song et al., 2020).[2] Appendix C provides more information on the distance weighting used. It also shows (Fig. 7) that in a more conventional setting, where a large manually labeled training set is available, the $k$-NN classifier clearly outperforms GPT-4 in accuracy (92% vs. 82%). Note that for the $k$-NN student, no retraining (Fig. 1) is necessary, since the cache coincides with the memory of the $k$-NN classifier. The cache is initialized with the 3-shot training examples of the classes (231 instances in total).

**Criteria:** We instantiate the criteria of Fig. 1 with two conditions. Both have to be satisfied for the student's response to be used; otherwise, we call the teacher. The first condition is that the cosine distance between the (MPNet-based) vector representation of the incoming message and the *weighted centroid vector* $\mathbf{c}$ of the $k$ nearest neighbors should be less than a threshold $t_c$. Here $\mathbf{c} = \sum_{i=1}^{k} \hat{w}_i \cdot \mathbf{v}_i$, and $\hat{w}_i = w_i / \sum_{j=1}^{k} w_j$, where $w_i$ is the weight assigned by distance weighting (Appendix C) to the $i$-th neighbour, and $\mathbf{v}_i$ is the (MPNet-based) vector representation of the neighbour. Intuitively, this condition ensures that the incoming instance is sufficiently close to cached instances.

To define the second condition, let $C$ be the set of the labels (classes) of the $k$ nearest neighbors (hereafter simply neighbors). Let $w_{i,c}$ be the weight (assigned by distance weighting) to the $i$-th neighbour belonging in class $c$, and let $W_c$ be the sum of all weights of neighbors of class $c$, i.e., $W_c = \sum_i w_{i,c}$.

We define the probability $p_c$ of each $c \in C$ as:

$$p_c = \frac{\exp(W_c)}{\sum_{c' \in C} \exp(W_{c'})}$$

The *entropy* $\mathcal{H}$ of the probabilities $p_c$ of the labels of the neighbors is:

$$\mathcal{H} = -\sum_{c \in C} p_c \log p_c.$$

The second criterion requires $\mathcal{H}_w$ to be less than a threshold $t_{\mathcal{H}}$. Intuitively, it requires the neighbors to agree on the label of the incoming instance.

**Hyperparameter tuning:** There are three hyperparameters here, the number of neighbors $k$, and the thresholds $t_c, t_{\mathcal{H}}$. We fix $k = 5$ as a practical choice considering that there are 3 examples per class initially. For each indicative $\lambda$ value (0.05, 0.1, 0.2, 0.3), we employ Bayesian optimization on the few-shot development set (Section 3) to determine the optimal combination of the two thresholds that maximize $\hat{\phi}$ (discounted accuracy). We let $t_c$ range in $[0, 2]$, and $t_{\mathcal{H}}$ in $[0, 4.34]$.[3] We use Optuna's (Akiba et al., 2019) implementation of the Tree-Structured Parzen Estimator (TSPE) algorithm (Bergstra et al., 2011) after first performing a $10 \times 10$ grid search on the range of values of the two thresholds as a head start. The resulting contour maps and the optimal values of the two thresholds per $\lambda$ value can be found in Appendix D.

**Results:** We evaluate OCaTS for each of the four indicative $\lambda$ values, using the same incoming instances (original test set of Banking 77), and the $\lambda$-specific tuned thresholds $t_c, t_{\mathcal{H}}$. As illustrated in Fig. 2, OCaTS succeeds in managing the tradeoff between calls to the teacher vs. accuracy. Figure 2 (left) shows that as the discount factor $\lambda$ increases, fewer calls to the teacher are made. In Fig. 2 (middle), we see how much accuracy is sacrificed for this OpEx relief. In particular, for $\lambda = 0.05$ the accuracy of OCaTS is very close to the accuracy of the GPT-4 teacher, within a margin of 0.37 percentage points (83.05% vs. 82.68% for the entire test set), while calling the teacher for only 1/3 of the incoming instances (1050 out of 3080). For higher values of $\lambda$, we see the intended drop in accuracy to achieve an increasingly smaller number of calls to the teacher. Figure 2 (right) shows that the discounted accuracy $\hat{\phi}$ of OCaTS (solid lines, one per

---

[2]We used `gpt-4-0314` and `all-mpnet-base-v2`, in particular, for the teacher and student, respectively.

[3]The maximum value of $\mathcal{H}$ with 77 classes is 4.34, when using natural logarithms. The upper bound of $t_c$ was chosen based on initial experiments on development data.

$\lambda$ value) is always clearly higher than the corresponding discounted accuracy of always calling the GPT-4 teacher (dashed lines). Hence, OCaTS is clearly better than always calling the teacher, if OpEx costs are taken into account. The difference increases (in favor of OCaTS) as $\lambda$ increases, i.e., as reducing OpEx costs becomes more important.

## 4 Conclusions

We introduced an Online Cost-aware Teacher-Student framework (OCaTS) to help SMEs reduce OpEx costs by caching the responses of commercial LLMs and training inexpensive local students. We also introduced discounted versions of common evaluation measures, allowing SMEs to quantify the trade-off between LLM calls and user frustration. By instantiating OCaTS with a GPT-4 teacher and a $k$-NN student and experimenting with an intent recognition dataset from the banking domain (Banking77), we showed that the calls to the teacher can be significantly reduced (by 1/3) with only a slight performance drop (0.37 percentage points). Additional experiments with an MLP student on the same dataset led to the same findings (Appendix E). Further experiments with a GPT-3.5 teacher, the initial $k$-NN student, and a sentiment analysis dataset (LMR) also confirmed the conclusions of the previous experiments (Appendix F).

In future work, we plan to experiment with more datasets and tasks (e.g., question answering), and suggest adaptive policies for $\lambda$ to allow higher OpEx costs (more frequent calls to the teacher) when the cache is cold and be more selective (calling the teacher less frequently) later on. We also plan to enhance OCaTS with indicators of how much we can trust the teacher responses (e.g., confidence of the teacher). Finally, we intend to incorporate more financial metrics (e.g., student costs) in the discounted versions of the evaluation measures and study more complex strategies (e.g., game-theoretic, reinforcement learning) to select the thresholds that determine when to trust the student or call the teacher.

## 5 Limitations

The main scope of this work was to propose a flexible framework (OCaTS) that will allow SMEs to reduce the OpEx costs when incorporating commercial LLMs in their solutions. We considered only two instantiations of the teacher (GPT-4, GPT-3.5) and two instantiations of the student ($k$-NN,

MLP) in two tasks (intent recognition, sentiment analysis), leaving further instantiations for future work. Although LLMs like GPT-4 and GPT-3.5 can in principle be used for zero-shot inference, we considered in-context learning with a few demonstrator examples per class. These examples where manually selected to be diverse and indicative of the corresponding classes. This is realistic to some extent; SMEs often request a small number of examples from their customers, but the quality of these examples is not always guaranteed. In addition, the test sets we used (from Banking77 and LMR) were balanced and thus not entirely realistic. However, we shuffle the stream of incoming (test) instances which, hence, do not arrive in a uniform way with the respect to their classes. Also, to tune $t_c$ and $t_{\mathcal{H}}$, we used a development set, extracted from the original training data. Such a development set is not always available in practice, but we used it for the sake of the analysis. Interested SMEs can use our analysis, as a starting point for their applications and reduce the number of trials needed to find suitable values for $t_c$ and $t_{\mathcal{H}}$.

Another limitation is that $\hat{\phi}$ takes into consideration only the cost to call the teacher ($\rho$), and indirectly the frustration of the user, as implied by the performance drop. A more detailed analysis would also incorporate the student cost and other financial metrics possibly with different weights; OCaTS can be easily extended in that direction. Finally, we did not compare against existing caching libraries, e.g., GPTCache.[4] These libraries are quite simplistic and less flexible than OCaTS, which can be used with a variety of teacher-student settings.

## 6 Ethics statement

Constantly querying LLMs to solve everyday tasks is not only costly; it has a large energy footprint as well. Our framework aims to alleviate both phenomena. Nonetheless, our study required a significant amount of resources. We believe, however, that by making the framework and the analysis publicly available, we can pave the way towards reducing the resources required by SMEs to handle their day-to-day tasks in the long run.

## Acknowledgements

This work was supported by Google's TPU Research Cloud (TRC) program.[5]

---

[4] https://github.com/zilliztech/GPTCache
[5] https://sites.research.google/trc/about/

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

# Appendix

## A   Statistics of the Banking77 dataset

Figure 3 shows the label distribution of the original training and test subsets of the Banking77 intent recognition dataset. The training subset exhibits a significant class imbalance (Fig. 3, left), whereas the test subset is balanced (right). In Table 1, we provide further statistics which, along with the label distribution, support the selection of the dataset as a realistic case to perform our experiments.

## B   More details about the LLM teachers

To prompt the LLMs, we used the chat completion API provided by OpenAI, which takes as input a

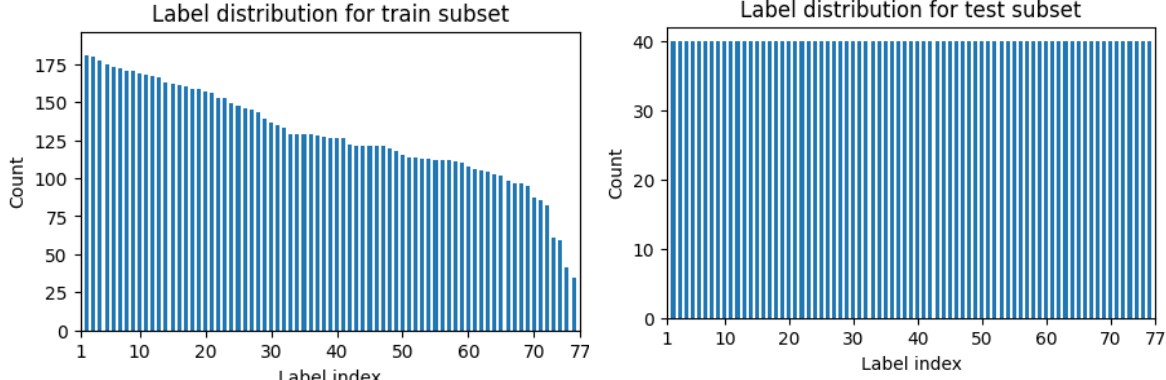

Figure 3: Label distribution of the original train (left) and test (right) subsets of Banking77.

| Statistics | Train | Test |
|---|---|---|
| Number of examples | 10,003 | 3,080 |
| Minimum length in characters | 13 | 13 |
| Average length in characters | 59.5 | 54.2 |
| Maximum length in characters | 433 | 368 |
| Minimum length in words | 2 | 2 |
| Average length in words | 11.9 | 10.9 |
| Maximum length in words | 79 | 69 |
| Number of intents | 77 | 77 |

Table 1: Statistics of Banking77.

```
You are an expert assistant in the field of
customer service. Your task is to help workers
in the customer service department of a company.
Your task is to classify the customer's question
in order to help the customer service worker to
answer the question. In order to help the worker
you MUST respond with the number and the name of
one of the following classes you know.
In case you reply with something else, you will
be penalized.
The classes are:
- activate_my_card
- age_limit
...
```

Figure 4: System message used in the GPT-4 teacher.

*system* message (instructions) along with a history of *user* and *assistant* messages, and generates an assistant message as output. The system message specifies the model's behavior as a chat assistant (Fig. 4). For our few-shot setting, we add to the system message a few pairs of user-assistant messages from the training set as demonstrators (Fig. 5). The incoming instance to be classified is added as the last user message in the history.

## C   Distance weighting in the k-NN student

The $k$-NN classifier assigns a weight to each one of the $k$ nearest neighbors of an incoming instance.

```
User: My new card is here, what's the process for
activating it?
Assistant: activate_my_card
User: I am unable to activate my card, it won't
let me.
Assistant: activate_my_card
User: Can you help me activate my card
Assistant: activate_my_card
User: What is the youngest age for an account?
Assistant: age_limit
User: What is the appropriate age for my child to
be able to open an account?
Assistant: age_limit
User: How do I set up an account for my children?
Assistant: age_limit
...
```

Figure 5: Demonstrators (in-context few-shot examples) used in the GPT-4 teacher as conversation history.

The weight is inversely proportional to the square of the cosine distance between the vectors of the incoming instance and the neighbor, $w_i = \frac{1}{d_i^2}$. The class with the largest sum of neighbor weights is then assigned to the incoming instance.

Figure 7 shows the learning curve of a distance-weighted $k$-NN classifier that is trained on the original training set of Banking77 and evaluated on the few-shot development set (Section 3). We observe that with sufficient training data (approx. 1000 instances) the $k$-NN classifier clearly outperforms GPT-4 in accuracy (92% vs. 82%), which justifies our choice to use it as a student in our experiments.

## D   Hyperparameter tuning

We tuned the thresholds $t_c$ and $t_{\mathcal{H}}$ anew for each $\lambda$ value. For each $\lambda$ value, we first performed a $10 \times 10$ grid search on the ranges of values of

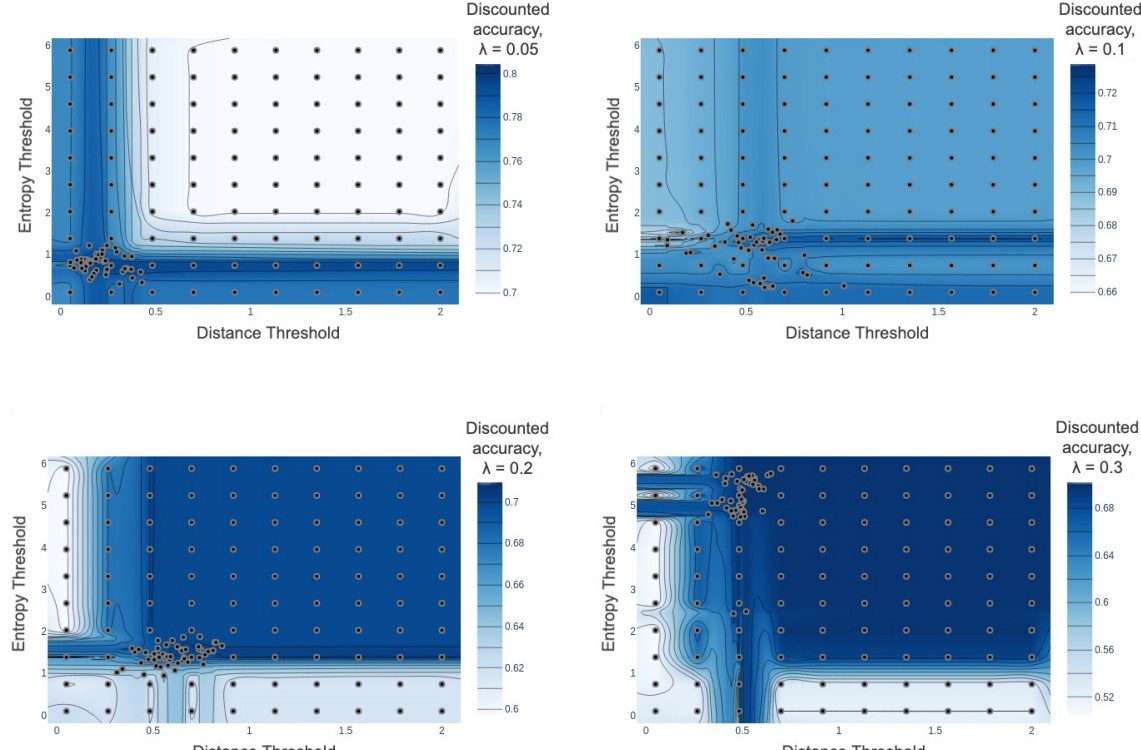

Figure 6: Contour plots (discounted accuracy) obtained during threshold tuning in the main experiments (GPT-4 teacher, $k$-NN student, Banking 77 data), for various $\lambda$ values.

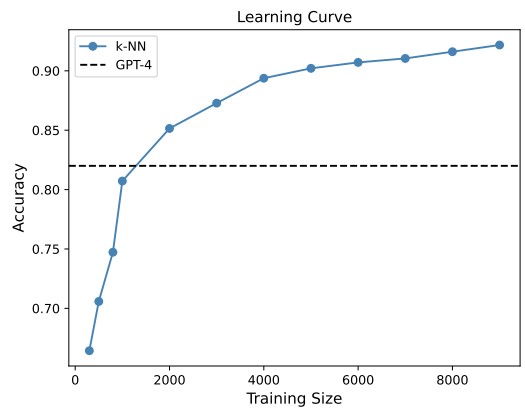

Figure 7: Learning curve of the distance-weighted $k$-NN student (solid line) when trained on the original training set of Banking77 and evaluated (accuracy) on the few-shot development set. Accuracy of GPT-4 on the few-shot development set also shown (dashed line).

the two thresholds, to evaluate indicative combinations for the thresholds. These are considered as starting points for the Bayesian optimization (Section 3). Figure 6 illustrates the presence of several good points adjacent to the best point in the main experiments (Section 3), all of which maximize the discounted metric to a significant extent. This

**System message:** Analyze the sentiment of the following reviews and classify them as either 'positive' or 'negative'.

**User:** When I started watching this movie I saw the dude from Buffy, Xander, and figured ah how nice that he's still making a living acting in movies. Now a weird movie I can stand, given that it's a good dose of weird like for example David Lynch movies, twin peaks, lost highway etc. [...] It wasn't his acting though, that was alright, but the script just didn't make any sense. Sorry.
**Assistant**: negative

**User**: As part of the celebration of the release of Casino Royale,v this film with the new Bond starring in it was shown, from director Roger Michell (Notting Hill). [...] Also starring Peter Vaughan as Toots, Danira Govich as Au Pair,Harry Michell as Harry, Rosie Michell as Rosie and Johnny English's Oliver Ford Davies as Bruce. Very good!
**Assistant**: positive
...

Figure 8: System message and demonstrators when using GPT-3.5 in the **sentiment analysis** task.

shows that several threshold combinations may be considered optimal. Moreover there are large areas often with a wide range of values for one threshold that are comparable in terms of maximizing the discounted accuracy measure. Table 2 provides the optimal threshold combination as selected by

the optimization process for each $\lambda$ value. We can observe that the tuned value for $t_c$ decreases from $\lambda = 0.2$ to $\lambda = 0.3$, instead of increasing, which can be accounted for by the aforementioned observation that multiple points maximize $\hat{\phi}$. However, we also notice the general increasing trend for both thresholds, which leads to fewer calls to the GPT-4 teacher, as one would expect, since higher $\lambda$ values mean that calls to the teacher are more costly.

| $\lambda$ | $t_{\mathcal{H}}$ | $t_c$ |
|------|--------|--------|
| 0.05 | 0.8359 | 0.2269 |
| 0.1  | 1.324  | 0.5656 |
| 0.2  | 1.558  | 0.76   |
| 0.3  | 2.336  | 0.4993 |

Table 2: Tuned thresholds per $\lambda$ in the main experiments (GPT-4 teacher, $k$-NN student, Banking77 dataset).

## E MLP student instead of k-NN student

As a first step to test the generality of the conclusions of our main experiments (Section 3), we repeated the main experiments this time using a Multi-layer Perceptron (MLP) student, instead of the $k$-NN student. The MLP has one hidden layer with a ReLU activation function, followed by a dropout layer, then an output layer with 77 neurons (number of classes) and a softmax. We use cross-entropy loss. Again, we initially filled the cache with 3 few-shot training instances per class ($3 \times 77 = 231$ in total) from the original training set (the same ones as in the main experiments), and the MLP was initially trained on them. We retrain the MLP every 100 calls to the teacher, i.e., whenever 100 new training instances have been added to the cache. We used multi-objective Bayesian optimization on the few-shot development set (treated as an incoming stream of user requests), optimizing for both loss and accuracy (no cost factors), to find the optimal hyperparameters for the MLP architecture. Subsequently, we tuned the threshold values ($t_c, t_{\mathcal{H}}$) for each lambda again using Bayesian optimization, as we did in the main experiments. The tuned hyperparameters and thresholds are shown in Table 3 and Table 4, respectively. For every incoming test instance, the entropy $\mathcal{H}$ (Section 3) is now computed using the output probabilities of the MLP. The other criterion (Fig. 1) remains unchanged, i.e., it requires the cosine distance of the MPNet-based vector of the incoming instance to the centroid of the $k$ most similar cached instances to be less than $t_c$; we use $k = 5$, as in the main experiments. As shown in Fig. 9, the experimental results with the

MLP student are very similar to those of the main experiments (cf. Fig. 7).

| Hyper-parameter | Range | Value |
|------|--------|--------|
| learning rate | $[10^{-5}, 0.1]$ | $1.6 \cdot 10^{-5}$ |
| hidden layer neurons | $\{256, ..., 1024\}$ | 1024 |
| dropout rate | $[0.1, 0.5]$ | 0.22 |

Table 3: Tuned hyper-parameters in the Banking77 experiments with the **MLP student** and GPT-4 teacher.

| $\lambda$ | $t_{\mathcal{H}}$ | $t_c$ |
|------|--------|--------|
| 0.05 | 0.48  | 0.896 |
| 0.1  | 0.479 | 1.553 |
| 0.2  | 0.583 | 1.464 |
| 0.3  | 0.646 | 1.882 |

Table 4: Tuned thresholds per $\lambda$ in the Banking77 experiments with the **MLP student** and GPT-4 teacher.

## F Sentiment analysis experiments

As a further step to confirm the conclusions of the previous experiments (Section 3, Appendix E), we conducted an additional set of experiments, now using a sentiment analysis task. We used the Large Movie Review (LMR) dataset (Maas et al., 2011), which consists of 50,000 IMDB reviews, and two labels (positive, negative review). Table 5 shows more statistics for this dataset.

Again, we assume that an SME can afford to create only a small number of training instances per class. To simulate this limited data setting, we created few-shot versions of the training and development sets of LMR, much as in Section 3. For the few-shot training set, we randomly selected 10 instances from the original training set, 5 for each sentiment, as follows. We employed the GPT-3.5 tokenizer from OpenAI[6] and computed the token size for each review. Subsequently, we randomly chose reviews with token sizes close to the median. Similarly, for the few-shot development set, we randomly selected 1,000 instances from the original training set, 500 for each sentiment. Finally, due to limited resources, we used only 5,000 instances

---

[6]https://github.com/openai/tiktoken

| Statistics | Train | Test |
|------|--------|--------|
| Number of examples | 25,000 | 25,000 |
| Minimum length in characters | 52 | 32 |
| Average length in characters | 1325.06 | 1293.7 |
| Maximum length in characters | 13704 | 12988 |
| Minimum length in words | 10 | 4 |
| Average length in words | 233.7 | 228.5 |
| Maximum length in words | 2470 | 2278 |
| Number of sentiments | 2 | 2 |

Table 5: Statistics of Large Movie Review (LMR).

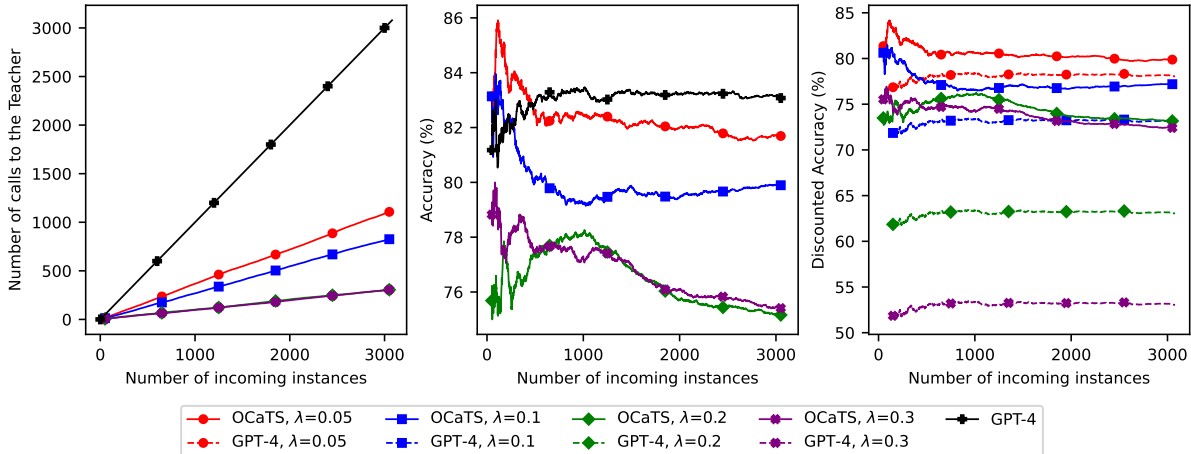

Figure 9: Number of calls to the teacher (left), accuracy (middle), and discounted accuracy (right), using a GPT-4 teacher and an **MLP student**, for various $\lambda$ values, on Banking77 data. In the left sub-figure, the green line (OCaTS, $\lambda = 0.2$) is not visible, because it overlaps with the purple one (OCaTS, $\lambda = 0.3$). The results are very similar to those of the main experiments (cf. Fig. 7). Again, OCaTS (right, solid lines) has better discounted accuracy than always calling the teacher (right, dashed lines) for all four indicative $\lambda$ values. The larger the $\lambda$, the fewer the calls to the teacher (left), at the expense of reduced accuracy (middle).

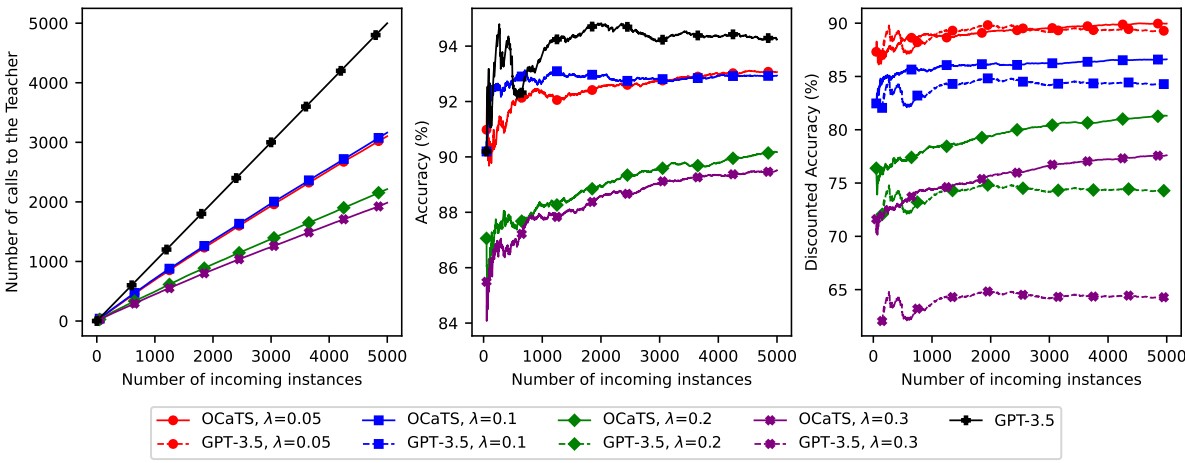

Figure 10: Number of calls to the teacher (left), accuracy (middle), and discounted accuracy (right), using a GPT-3.5 teacher and a $k$-NN student, for various $\lambda$ values, on **sentiment analysis** (LMR) data. The results are very similar to those of the previous experiments (cf. Figures 2 and 9).

of the original test set as the stream of incoming instances. As in the previous experiments, we repeat each experiment with five random shufflings of the incoming stream, and report the average scores. We use the distance weighted $k$-NN classifier, as in Section 3. Again, we set $k = 5$, and we employ Bayesian optimization on the few-shot development set to determine the optimal combination of the two thresholds that maximize $\hat{\phi}$. We let $t_c$ range in $[0, 2]$, and $t_{\mathcal{H}}$ in $[0, 0.7]$.[7] For the teacher, we used the cheaper GPT-3.5 in these experiments.[8] We prompted GPT-3.5 using the same in-context learning approach outlined in Section 3; we provided the few-shot training set we created as demonstrators and asked GPT-3.5 to classify the review as either positive or negative (Figure 8). Figure 10 shows that the results were very similar to those of Section 3 and Appendix E.

---

[7] The maximum value of $\mathcal{H}$ with 2 classes is 0.69 when using the natural logarithm.

[8] We used version `gpt-3.5-turbo-0301`.