# OpenReview forum: "Cache me if you Can: an Online Cost-aware Teacher-Student framework to Reduce the Calls to Large Language Models"
_EMNLP/2023/Conference — EMNLP 2023 Findings_

### Official Review · Reviewer_4DeH · 2023-08-04

**Soundness:** 4

**Excitement:**

4: Strong: This paper deepens the understanding of some phenomenon or lowers the barriers to an existing research direction.

**Paper Topic And Main Contributions:**

Authors propose a simple yet effective approach to decide when to make an expensive call LLM call vs when can past knowledge be used by caching and avoid the LLM call. Authors propose a teacher-student framework based on kNN classifier. Authors also introduce a discounted evaluation measure. Authors evaluate their approach and provide initial baselines on employ Banking77 dataset.

**Questions For The Authors:**

* Could you add stats around what % of request both in train and test have same responses.
* What is the rationale behind using a discounting factor in the eval metric?
* It is not clear how the current algorithm handles a new label. Could you elaborate on that in the paper?

**Reasons To Accept:**

* The proposed teacher-student framework is a simple yet effective solution for teams trying to optimize for cost and latency in using expensive LLM calls.

**Reasons To Reject:**

* The current framework seems it cannot handle compositional or compound queries where a correct response to a subtask is present in the cache but the other requires LLM call.
* It is not clear how the current algorithm handles a new label that was not part of training data.

**Reproducibility:**

1: Could not reproduce the results here no matter how hard they tried.

**Reviewer Confidence:**

5: Positive that my evaluation is correct. I read the paper very carefully and I am very familiar with related work.

---

> ### Author Rebuttal · Authors · 2023-08-28
>
> Thank you for your valuable feedback.
>
> Indeed, in more complex compositional or complex queries one may need to use only parts of cached instances and query the LLM for other parts. We would like to point out, however, that this is a short paper and in the limited available space we present a new  problem (reducing the calls to costly LLMs by using a local student), a general framework for addressing it (OCaTS), and a first incarnation/evaluation of the framework in a particular task (intent recognition). We plan to explore more complex tasks, including compound queries, in the future, and we also expect that others will want to explore and extend our framework in a variety of tasks. Please see also our related response to Reviewer 5m6Q. We will update the manuscript accordingly to acknowledge these points.
>
> About handling new labels, we would like to note that this is a first instantiation of OCaTS and we opted for a simple setting to evaluate the framework. Handling cases where the label set changes (e.g., by adding new labels or deleting existing ones) is a very practical problem in commercial settings and a line of research on its own;  we plan to address this interesting direction in future work and we will update the manuscript accordingly to acknowledge this point.
>
> About the percentage of requests that have the same response, note that we experiment with a single-label intent classification task with 77 intents. Given a user message, the response is the intent of the user. From the training data, we use only 231 examples (exactly 3 per class) that are used both for the teacher to perform in-context learning (as demonstrators in the prompt) and as the initial cache of the student (warm up). Consequently, for each incoming test instance there will be exactly 3 instances of the test intent in the context of the teacher LLM, and at least 3 instances of the test intent in the student’s cache; the cache starts at a balanced state, but it may evolve to less balanced states if the LLM happens to be queried more frequently for some classes (intents). We can include figures showing the evolution of the class distribution in the camera-ready.
>
> Regarding the discounting factor, increasing the accuracy usually comes at a cost, i.e., making a paid call to the teacher model. On the other hand, using the cost-effective student model leads to lower accuracy. The discounted evaluation measure is a way to quantify the trade-off in a single value, allowing to compare different instantiations of the teacher-student framework. We discuss this in Section 2 (lines 154-162), but we will make the discussion clearer in the final manuscript, also adding examples of cost savings in dollars (as suggested by reviewer 7Vyf) to better ground the discussion.
>
> About reproducibility, we do use a public dataset (BANKING77) and provide in the appendix all the hyper-parameters and prompts required to repeat our experiments. We also plan to release the code upon acceptance.

---

### Official Review · Reviewer_7Vyf · 2023-08-05

**Typos Grammar Style And Presentation Improvements:** Figure 2 - Discounged
**Soundness:** 4

**Excitement:**

3: Ambivalent: It has merits (e.g., it reports state-of-the-art results, the idea is nice), but there are key weaknesses (e.g., it describes incremental work), and it can significantly benefit from another round of revision. However, I won't object to accepting it if my co-reviewers champion it.

**Missing References:**

These are a few papers that study the related problems of delegating and abstaining classifiers as well as other cost-reducing approaches like early exiting:
- César Ferri, Peter Flach, and José Hernández-Orallo. 2004. Delegating classifiers. In Proceedings of the twenty-first international conference on Machine learning (ICML '04). Association for Computing Machinery, New York, NY, USA, 37. https://doi.org/10.1145/1015330.1015395
- Tadeusz Pietraszek. 2005. Optimizing abstaining classifiers using ROC analysis. In Proceedings of the 22nd international conference on Machine learning (ICML '05). Association for Computing Machinery, New York, NY, USA, 665–672. https://doi.org/10.1145/1102351.1102435
- B. Barla Cambazoglu, Hugo Zaragoza, Olivier Chapelle, Jiang Chen, Ciya Liao, Zhaohui Zheng, and Jon Degenhardt. 2010. Early exit optimizations for additive machine learned ranking systems. In Proceedings of the third ACM international conference on Web search and data mining (WSDM '10). Association for Computing Machinery, New York, NY, USA, 411–420. https://doi.org/10.1145/1718487.1718538

I think these would be good starting points for a related work section.

**Paper Topic And Main Contributions:**

This paper studies the topic of mitigating the cost of LLM inference by delegating simple examples to an inexpensive preliminary classifier.
The primary contribution is an approach for using the LLM predictions to improve the preliminary classifier in an online setting.
The approach employs an embedding-based kNN classifier over past LLM predictions as the preliminary classifier.
The decision of whether to delegate to the LLM is based on whether the distance between the input embedding and the embeddings of the nearest neighbors exceed a learned threshold.
The decision threshold is learned to optimize a discounted objective function (e.g., accuracy) where the discount factor can be tuned to weigh the relative cost of performing inference with the LLM vs. making a mistake.
The paper also contributes experimental results applying the method to the Banking77 intent classification datasets.
The experimental results demonstrate that the proposed system can achieve accuracy within 0.5% (absolute) of the LLM, while only using the LLM to make predictions on 1/3 the number of examples.

**Questions For The Authors:**

You write that:
> λ can intuitively be thought of as a currency exchange rate, showing how expensive ρ is in terms of δ (e.g., loss of accuracy in percentage points).

I think that this is an interesting point, but it would be even more interesting if it were grounded in terms of real costs.
I have a few questions about this:

d. In your setting would it be possible to back this out to an estimate of the dollar cost of an additional point of accuracy?

e. How would this compare to the dollar cost of an additional point of accuracy from using human-annotated labels instead of LLM generated labels?

I understand that the second question may be difficult to answer without knowing the cost of collecting Banking77 (it looks like the experimental portion is covered by the results in Figure 3), but if you were able to quantify whether/when an additional point of accuracy is using LLMs in your framework, I think this paper would be much more exciting.


**Reasons To Accept:**

a. Performing inference using proprietary LLMs is becoming a substantial operating expense for businesses as well as NLP research labs. The proposed method could be helpful in reducing this cost while maintaining comparable levels of accuracy.

**Reasons To Reject:**

b. Novelty. The idea of having the preliminary classifier abstain from making predictions (i.e., delegate making predictions to the LLM) resembles a large body of existing work on abstaining/delegating classifiers and early exiting.
   I believe that the novel aspect of this work is that an LLM is being used to generate the labels instead of a human labeler, which, given the current popularity of LLMs, is in my opinion a relatively straightforward modification.
   This accounts for the majority of my excitement score; if there is a novel aspect of the paper that I am missing I would be willing to revise this score upwards.
c. Limited Scope. Experimental results are only reported for a single task, thus it is hard to generalize how well this method may work in other settings. This negatively impacts my evaluation of the paper's soundness.

**Reproducibility:**

4: Could mostly reproduce the results, but there may be some variation because of sample variance or minor variations in their interpretation of the protocol or method.

**Reviewer Confidence:**

3: Pretty sure, but there's a chance I missed something. Although I have a good feel for this area in general, I did not carefully check the paper's details, e.g., the math, experimental design, or novelty.

---

> ### Author Rebuttal · Authors · 2023-08-28
>
> Thank you for your valuable feedback. Please find below our response to your points (b)-(e).
>
> (b) We thank the reviewer for pointing us to abstaining and delegating classifiers and early exit. Unlike abstaining classifiers, which do not provide classification decisions when they are not confident enough, our Teacher-Student model always provides decisions. In that sense, our work is closer to delegating classifiers. In particular, our Student is similar to the cautious first-level classifier of two-level delegating classifiers. However, our Teacher is not trained, unlike the second-level classifiers of delegating classifiers. More generally, delegating classifiers were intended to be more efficient alternatives to classical ensembles (e.g., bagging or boosting), sharing with them the main goal of improving classification performance (e.g., accuracy). In our case, the goal is to save money, by calling less frequently an already high-performance, but costly and not further trained Teacher LLM, and accepting some performance loss (due to the simpler, but cheaper Student). In that sense, our work is related to early-exit approaches, which use a light/cheap first model to filter out easy instances that do not need to go through a more elaborate and expensive second model. However, we are not aware of any previous early-exit work that aims to reduce the calls to expensive LLMs. As in delegating and some early-exit classifiers, we tune the delegation/exit threshold (actually two thresholds in our k-NN Student, based on the distance and entropy of the neighbors), but to minimize discounted-accuracy, which considers both the accuracy of the overall Teacher-Student system and the cost of invoking the Teacher. So, the goal, setup, and objective function that we use are quite different compared to abstaining, delegating, and previous early-exit classifiers. These points can be easily added to the paper, along with the suggested references.
>
> (c) We would like to point out that this is a short paper, and in the limited space available we present a new timely problem (reducing the calls to costly LLMs by using a local student), a general framework for addressing it (OCaTS), and a first incarnation/evaluation of the framework in a particular task (intent recognition). Please see also our related response to Reviewer 5m6Q.
>
> (d) We have used a percentage of calls to the LLM because the cost depends on the actual rate of user messages per day or per month. For example, for 10k user messages to be classified, using the tuned hyperparameters we can project that 2/3 of calls to the LLM will be omitted, saving essentially 2/3 of \$ 2,400 per day (see note below), i.e., \$ 1,600 per day for a negligible drop in accuracy of 0.37 p.p. (Note that a 3-shot approach with 77 classes results in a context of approx. 8k tokens per classification call to GPT-4, costing approx. 8 x \$ 0.03 = \$ 0.24 per classification call). This example shows that we can retain the LLM accuracy while saving significant operational costs. Similar discussion of indicative cost savings in dollars will be added to the paper and we thank the reviewer for the suggestion.
>
> (e) We would like to stress that OCaTS aims to address an online setting, where there is no prior comprehensive dataset and instances come from the users at the time the instances need to be handled. (Such an online setting is common in business practice, when solutions for new clients need to be developed quickly.) Although we could estimate the costs of a human annotator, e.g., with a naive approach by means of their salary, we note that employing a human annotator is not a viable alternative for a commercial online setting, where each user message needs to be handled within milliseconds. Alternative setups can be devised where a human annotator is periodically employed to annotate messages that the teacher model has already handled, but this is out of scope for this short paper aiming at proposing OCaTS as a online cost-aware teacher-student framework. We will, nevertheless, add related discussion to clarify this point.

---

### Official Review · Reviewer_5m6Q · 2023-08-11

**Soundness:** 3

**Excitement:**

2: Mediocre: This paper makes marginal contributions (vs non-contemporaneous work), so I would rather not see it in the conference.

**Paper Topic And Main Contributions:**

The paper proposes techniques to cache the queries to reduce operational costs at the expense of reduced performance.

**Reasons To Accept:**


+ Investigation of a timely problem.
+ Reduces cost for enterprises.


**Reasons To Reject:**

- Weak evaluation, only a single dataset, and limited NLP tasks (intent recognition). The paper would improve if additional NLP tasks (sentiment classification, and named-entity recognition, among others) were considered.
- Uses proprietary GPT-4 as a teacher, which may not be cost-effective. Open-source variants make more sense if the goal is to save costs.
- No clear description of when to flush and update the cache. For instance, the cached version may contain older data. How often will the model fetch? What are the performance trade-offs?

**Reproducibility:**

2: Would be hard pressed to reproduce the results. The contribution depends on data that are simply not available outside the author's institution or consortium; not enough details are provided.

**Reviewer Confidence:**

2: Willing to defend my evaluation, but it is fairly likely that I missed some details, didn't understand some central points, or can't be sure about the novelty of the work.

---

> ### Author Rebuttal · Authors · 2023-08-28
>
> Thank you for your valuable feedback.
>
> Indeed it is a limitation that we use a single dataset for evaluation, and we will update the limitations section accordingly. Having said that, we would like to point out that this is a short paper, and in the limited space available we present a new problem (reducing the calls to costly LLMs by using a local student), a general framework for addressing it (OCaTS), and a first incarnation/evaluation of the framework in a particular task (intent recognition). We believe that making this work publicly available will foster research on a timely (as noted by the reviewer) problem and will allow others to experiment with our framework in a variety of other tasks, significantly reducing costs for enterprises (as also noted by the reviewer).
>
> About using open source models, we would like to point out that deploying LLMs to support commercial services in production requires significant operational costs, including expensive infrastructure and a DevOPs/MLOPs team to coordinate resources in order to handle a high throughput (e.g., of hundreds of users per minute). Therefore using open source LLMs is not necessarily more cost-effective in commercial settings, and requires further analysis that we leave for future work. We will update the manuscript accordingly to clarify this aspect.
>
> About selecting GPT-4 as a teacher, we have explored less expensive commercial LLMs, including GPT-3.5 and Claude, and they significantly lag behind in micro-F1 (7.9-9.3 p.p.).  We will clarify this in the manuscript accordingly.
>
> About flushing the cache, retraining the student (which includes selecting which accumulated data to use/discard) is included in the general framework (see Fig. 1 and Section 2), but further analysis of this aspect is outside the scope of this short paper that aims to present the problem, the general framework for addressing it, and a first incarnation/evaluation of the framework. We will more clearly acknowledge this point in the limitations section.
>
> About reproducibility, we do use a public dataset (BANKING77) and provide in the appendix all the hyper-parameters and prompts required to repeat our experiments. We also plan to release the code upon acceptance.

---

### Meta-Review · Area_Chair_Pyu6 · 2023-09-17

**Recommendation:** 4

**Metareview:**

The paper addresses a new problem of high practical relevance in education, namely how the costs of calls to LLMs can be reduced by using previous results of LLM calls in a knn fashion. The paper proposes a simple yet effective solution to a timely problem which seems appropriate for a short paper, although reviewers criticize some technical details (how to update the cache exactly, how to handle new labels) and a somewhat limited evaluation on a single dataset only.

---

### Decision · Program_Chairs · 2023-10-07

**Decision:**

Accept-Findings

**Comment:**

The paper addresses a new problem of high practical relevance in education, namely how the costs of calls to LLMs can be reduced by using previous results of LLM calls in a knn fashion. The paper proposes a simple yet effective solution to a timely problem which seems appropriate for a short paper, although reviewers criticize some technical details (how to update the cache exactly, how to handle new labels) and a somewhat limited evaluation on a single dataset only.